# Changes in the Average Body Mass Index of Fifth- and Sixth-Grade Korean Elementary School Students: The Role of Physical Education in Student Health

**DOI:** 10.3390/healthcare12080855

**Published:** 2024-04-18

**Authors:** Byung-Kweon Chang, Se-Won Park, Young-Sik Kim, Seung-Man Lee

**Affiliations:** 1Department of Physical Education, Korea National University of Education, Cheongju 28173, Republic of Korea; jangbk82@gmail.com (B.-K.C.); sewonxxx@gmail.com (S.-W.P.); 2Department of Sports Science, Hankyong National University, Anseong 17579, Republic of Korea

**Keywords:** BMI, physical education, health, elementary school

## Abstract

This study aimed to emphasize the importance of physical education in maintaining sufficient physical activity by examining the average BMI of fifth- and sixth-grade elementary school students in Korea over multiple years. This study investigated changes in the average BMI of fifth- and sixth-grade elementary school students in Korea. It provided foundational data to suggest the role of physical education in student health and set future directions. The 2019–2021 Physical Activity Promotion System (PAPS) of the Korean Ministry of Education was used. Data corresponding to fifth- and sixth-grade students (124,693 from 2019, 126,226 from 2020, and 117,809 from 2021) in 1208 elementary schools in Korea were collected. Repeated measures ANOVA was conducted for the data analysis. The study findings are as follows: First, the average BMI of fifth- and sixth-grade elementary school students in Korea increased significantly in 2021 compared with 2019 and 2020. Second, changes in the yearly average BMI showed a significant difference depending on gender and grade. Obesity among fifth- and sixth-grade elementary school students in Korea increased steadily during 2019–2021, which may be due to a decrease in students’ physical activity. Male students showed a steeper increase regardless of grade. During the study period, limited physical activity at school increased obesity rates. Physical education must play a role in students’ health in preparation for future changes.

## 1. Introduction

Physical education has recently faced a remarkable change, which was especially promoted by the 2020 COVID-19 pandemic. Globally, trends such as the rapid spread of physical activity using non-face-to-face remote systems [1,2], the verification of health promotion and treatment effects [3,4], and the convergence of physical activity and information and communications technology (ICT) have been observed [5,6,7]. After the pandemic, a new popular trend called “EdTech” appeared [8,9,10].

Despite the positive aspects of this change, people inevitably faced negative aspects. The biggest drawback is that there is a lack of a sufficient amount of physical activity [11]. Insufficient physical activity has caused several problems in various subjects. Notably, the increase in obesity, which threatens student health, has been reported in several studies [1,12,13]. While these studies consistently focused on physical activity and obesity, they did not examine the changes in obesity levels in a large student population at the national level, contemplating the role of physical education for student health from a pedagogical perspective. Schools have the positive aspect of preventing overweight and obesity because they can universally provide programs that guarantee a certain level of health and physical activity for students.

Body mass index (BMI), which is the most common index for determining obesity, is calculated by dividing weight by the square of height (kg/m^2^) [14]. Since the BMI can be measured using only height and weight, it has been widely used as an efficient value for measuring obesity. Moreover, a relatively long argument is being carried out on the necessity of measuring children’s BMI and providing the results to parents during school years [15]. Physical education provided at school is known to have a positive effect on obesity prevention [16,17]. This is further supported by research results indicating that the incidence of obesity in children increased due to school closures during the COVID-19 pandemic [18]. Additionally, school physical education has been reported to have effects such as improving physical strength and fostering character development [19,20].

This study tracked changes in obesity among Korean elementary school students using large-scale national-level data. This is significant in that it targeted a repeated and large number of subjects that could not be included in previous studies. This research is also necessary as it can provide basic data for setting the direction of national policies and future education. Furthermore, the period in which this study was conducted was when non-face-to-face physical education classes were actively conducted. This study explored changes in obesity during this period to examine the effect of changes in physical education. Furthermore, this was conducted to contemplate the present role and the future direction of physical education.

Therefore, this study sought to examine how BMI values in Korean elementary schools have changed over the years using a large-scale survey measured repeatedly for 3 years, through which it will provide basic data for diagnosing the role of physical education for student health and setting future directions. For this purpose, the following research questions were formulated.

First, how did the average BMI of fifth- and sixth-grade elementary school students in Korea change according to each year of the study period?

Second, how did the yearly average BMI of fifth- and sixth-grade elementary school students in Korea change depending on gender?

Third, how did the yearly average BMI of fifth- and sixth-grade elementary school students in Korea change depending on the grade?

## 2. Materials and Methods

This study aimed to explore the role of physical education in student health by examining the changes in the average BMI of 5th- and 6th-grade elementary school students in Korea. To this end, this study used the BMI data of Korean elementary school students. BMI was measured as a subfactor of the Physical Activity Promotion System (PAPS) implemented according to the plan established by the Korean Ministry of Education [21]. The PAPS is a “student health physical fitness evaluation system” implemented every year on 5th- and 6th-grade elementary school students, as well as on all middle school (1st to 3rd year) and high school students (1st to 3rd year) nationwide. Measurements of cardiorespiratory endurance, flexibility, quickness, body fat, and muscular strength/endurance are mandatory, while precise cardiorespiratory endurance, obesity, and student posture assessments, as well as body self-assessment, are optional [22]. BMI is measured as a mandatory element for weight status. The yearly results of the PAPS for elementary, middle, and high schools nationwide are disclosed on the school information disclosure service School Info (https://www.schoolinfo.go.kr, accessed on 17 January 2020.), particularly the male and female BMI averages according to grade. The 2019–2021 PAPS results were disclosed on School Info as of January 2023. It has been used in various studies on student health because it is a highly reliable large-scale survey conducted by a public educational institution [21,23].

### 2.1. Participants

The initial sample population of this study consisted of all 6309 elementary schools in Korea (as of May 2021). After excluding those with no measured values in even one of the four groups (5th-grade male students, 5th-grade female students, 6th-grade male students, 6th-grade female students) for reasons such as not implementing the PAPS due to the COVID-19 pandemic between 2019 and 2021, a total of 1208 elementary schools remained in the final sample. In 2020–2021, the Korean Ministry of Education allowed schools to autonomously decide whether and when to measure the PAPS, which had been measured every year until then [21,22]. The number of students belonging to these schools was 124,693 in 2019, 126,226 in 2020, and 117,809 in 2021; the characteristics of the subjects are shown in Table 1. This study was approved by the Institutional Review Board of the Korea National University of Education (KNUE-202403-SB-0110-01).

### 2.2. Items and Measurements

In this study, BMI was used as the measurement tool to compare weight status depending on year, gender, and grade. It was measured to determine obesity and calculated by dividing weight by the square of height (kg/m^2^) [24]. Elementary schools in Korea enter students’ height and weight into the National Education Information System (NEIS), where the BMI is automatically calculated. Measurements are made at the school clinic by a certified school nurse. The final measurements are provided to students and parents mandatorily after being printed out individually. The standard table for the BMI of elementary school students set by the Korean Ministry of Education is shown in Table 2.

### 2.3. Procedure and Statistical Analysis

This study used the following procedures to process data. First, frequency analysis was conducted to determine the demographic characteristics of the subjects. Second, repeated measures ANOVA was conducted to explore changes in the average BMI. Changes in the average depending on year and gender and year and grade were also explored, in addition to the year, as variables. The statistical significance level was set at *p* < 0.05, and confidence intervals were modified using the Bonferroni correction for multiple comparisons [21]. As a result of the sphericity test in the process, the interaction effect between average BMI and year failed to meet the sphericity assumption (*p* < 0.05), and thus the Greenhouse–Geisser correction was used for the effect test [24]. Statistical significance was set at α = 0.05. Data were analyzed using SPSS version 18.0 (IBM Corp., Armonk, NY, USA).

## 3. Results

### 3.1. Average MBI Depending on Year, Gender, and Grade

The results showed that the main effect of the year was significant on the average BMI (*p* < 0.001). Moreover, the interaction effect between gender and year (*p* < 0.001) and between grade and year (*p* < 0.01) was also significant. However, the interaction effect between gender, grade, and year was not significant. In other words, the average BMI changed significantly depending on year, gender/year, and grade/year (Table 3).

#### 3.1.1. Average BMI by Year

The results of the changes in the average BMI depending on the year are shown in Table 4. The average BMI increased in 2021 (M = 21.241) compared with 2019 (M = 20.325) and 2020 (M = 20.394; *p* = 0.000). While an increase was observed in 2020 compared with 2019, it was not statistically significant.

#### 3.1.2. Average BMI According to Gender and Year

The results of changes in the average depending on gender and year are shown in Table 5 and Figure 1. Male students (M = 20.870) showed a higher average BMI than female students (M = 19.780) in 2019 (*p* = 0.000). The gap between male (M = 20.952) and female students (M = 19.837) increased in 2020 compared with 2019 (*p* = 0.000), reaching the largest gap in 2021 (M = 22.012 and M = 20,470, respectively; *p* = 0.000).

#### 3.1.3. Average BMI According to Grade and Year

The results of changes in the average BMI by grade and year are shown in Table 6 and Figure 2. In 2019, the average BMI of the sixth grade (M = 20.659) was higher than that of the fifth grade (M = 19.991; *p* = 0.000). In 2020, the gap between grades increased compared with 2019 (M = 20.797 and M = 19.992, respectively; *p* = 0.000). However, in 2021, the gap between the sixth (M = 21.497) and fifth (M = 20.985) grades decreased, showing the smallest gap in 3 years (*p* = 0.000).

## 4. Discussion

This study investigated changes in the average BMI of a large population consisting of fifth- and sixth-grade elementary school students in Korea over 3 years (2019–2021). Ultimately, this study sought to diagnose the role of physical education and provide basic data to explore the direction to prepare for future society. Accordingly, the discussions are as follows.

### 4.1. Interpretation of Findings

#### 4.1.1. Changes in the Average BMI by Year

According to the average BMI results by year, there was no change in 2020 compared with 2019, but there was a significant increase in 2021. Previous studies clarified that the cause of increased obesity lies in decreased physical activity. A comparative study on obese and non-obese children from racially diverse middle school students reported that lack of physical activity was an important factor in maintaining childhood obesity [25]. There was a negative view of the approach that focused on a healthy diet and physical activity as solutions to overweight and obesity; however, a summary of studies conducted thus far claimed that physical activity and exercise still had potential as solutions to obesity [26]. Some studies mentioned that the increase in ICT use was a cause of the increase in obesity. It was reported that Finnish adolescents’ use of ICT, such as watching TV, playing digital games, and using computers, was related to the prevalence of overweight and obesity [12]. To prevent this, adolescents’ exposure to electronic devices must be regulated by their parents [27].

Notably, the results of this study showed that the BMI increased significantly from 2020 to 2021 compared with from 2019 to 2020. The increase in body fat and decrease in physical fitness showed similar patterns during the same period in previous studies [21,28], and the decrease in physical activity was pointed out as the cause. At the time, the decrease in physical activity was due to the social distancing measures and the increase in non-face-to-face learning because of the pandemic, which is one of the options that can be chosen by physical education in the big trend of future education.

The active application of non-face-to-face education and EdTech is an unavoidable trend in future education. However, this study clearly revealed that physical education fails to fulfill its role in these environmental changes and that limiting students’ physical activity may lead to an increase in obesity. Furthermore, since a lack of physical activity contributes to various chronic diseases and health complications, it is necessary to increase the amount of physical activity for students at the national level [29]. In particular, active participation in various forms of physical activity in childhood is an important prerequisite for participation in physical activity in adulthood [30]. Thus, physical education in public education must establish measures for these changes in a way that can resolve the decline in students’ physical fitness and the increase in obesity.

In other words, physical education in schools of the future must be the last resort to ensure physical activity among students. The role of physical education is to make up for the insufficient amount of physical activity given to students both inside and outside of school. Efforts must be made in physical education to develop teaching and learning methods that can guarantee or supplement a sufficient amount of physical activity while actively using EdTech.

#### 4.1.2. Changes in the Average BMI Based on Gender and Year

The yearly average BMI results showed differences by gender. This difference gradually increased, reaching its peak in 2021. Furthermore, male students showed a greater increase in BMI than female students. These results are in line with a study [21] showing that the physical fitness level, including body fat, of Korean male elementary school students deteriorated more than that of female students during the same period as this study. In addition, many studies reported that limited physical activity had a more negative impact on men than on women [31]. This is because male students engage in more physical activity than female students [32]; therefore, the restrictions on physical activity due to COVID-19 had a greater negative impact on male students than female students. According to a study exploring data from the National Health and Nutrition Examination Survey (NHANES) in the United States from 1999 to 2006 among adolescents aged 12 to 19, 16.6% of male and 15.3% of female adolescents participating in recreational sports were obese (based on BMI) [33]. However, male adolescents not participating in leisure sports showed an obesity rate of 23.6%, whereas female adolescents showed one of 17.0%. In other words, male adolescents showed a greater difference in BMI when not participating in physical activity.

There are also results that are contrary to those of this study. According to previous studies related to ICT use among Finnish adolescents (ages 14, 16, 18, N = 6615), female students showed a higher prevalence of overweight and obesity (based on BMI) than male students from watching TV and using computers [12]. The results imply that male students tend to play more computer games than watch TV compared with female students, and as computer games have also recently involved physical activity, it may affect the amount of physical activity depending on the type of ICT they use.

#### 4.1.3. Changes in the Yearly Average BMI by Grade

The average BMI results showed yearly differences depending on grade. While this difference increased slightly in 2020 compared with 2019, it decreased in 2021, showing the smallest difference in three years. This implies that an increase in the period in which physical activity is restricted leads to an increase in BMI regardless of grade and age.

Previous studies have suggested various variables that affect obesity. According to a study examining adults in 15 European Union (EU) countries, the prevalence of obesity was higher in men than in women and in those with lower levels of education. Moreover, less participation in leisure or physical activity led to less interest in physical activity participation, and more time spent sitting at work led to higher obesity rates; further, married couples, widows, and divorced individuals showed a higher prevalence than singles [34]. Meanwhile, one notable variable in the adolescent group was sleep quality [35]. According to this study exploring the relationship between sleep quality and obesity among male and female adolescents (ages 11–16), obese adolescents showed shorter sleep duration than non-obese adolescents. Sleep disorders were not correlated with obesity but affected the level of physical activity.

Many previous studies proved that physical activity is a common factor that affects obesity. Although diverse variables may affect obesity, it can be expected that these variables are related to the amount or level of physical activity. In this study, a difference in BMI was observed depending on the grade during the first half of the research period (2019–2020); however, this difference decreased in the latter half (2020–2021), indicating that the grade did not play a significant role.

### 4.2. Practical Implications of This Study

In summary, the changes in the educational environment during the survey period (2019–2021) reduced physical activity, which increased the average BMI of Korean elementary school students. This was a reminder of how important school physical activity is, and it raised the need to use ICT in a way that involves physical activity in future education [36]. In particular, the increase in BMI stood out more among male than female students, and the effect depending on grade decreased.

Based on the above, the following suggestions can be made for the role and policy of physical education to prevent a decrease in physical activity due to changes in the future environment.

First, it is necessary to improve the quality and quantity of physical education programs to create a healthy lifestyle for students. Physical education is responsible for nurturing healthy and vibrant citizens [29], and since these habits are formed from childhood to adolescence when physical education is introduced [37], students must be provided with opportunities to participate in physical activity and feel the joy of it so that they can engage in it more [38].

Second, it is necessary to prepare for physical activity even in non-face-to-face situations if face-to-face physical education cannot be provided due to various environmental constraints. While students can engage in joint physical activity at school, they can engage in individual physical activity at home [39]. For example, it will be possible to meet the required amount of physical activity for adolescents by measuring and managing the number of steps and amount of exercise using ICT devices and doing exercises or dancing for physical fitness using home games.

Third, parents must pay more attention to their children’s physical activity at home and actively ensure the necessary amount of physical activity for students. As found in previous studies, adolescence is a time when physical activity habits are formed, which is why the role of parents and constant recommendations of physical activity is important so that students can engage in it even at home [40].

### 4.3. Limitations and Scope for Further Research

Several limitations were found in the process of producing these results, and suggestions for future studies center on these limitations. First, this study was a large-scale project targeting all fifth- and sixth-grade elementary school students in Korea. Further research should use and compare data by continent, country, and race. Second, this study derived objective results using large-scale quantitative data. By understanding the individual context through a qualitative method, it would be possible to gain an in-depth understanding of the impact on the physical activity and health of each individual. Last, this study was conducted to explore the role of physical education by examining the changes in the average BMI of elementary school students in the context of a decline in physical activity. The findings of this study are merely a starting point for reestablishing the role of physical education in future society. Therefore, further research should be conducted to develop a specific program through which physical education can guarantee a sufficient amount of physical activity for students in future schools.

## 5. Conclusions

This study investigated changes in the average BMI in 1208 Korean elementary schools using the PAPS data measured in 2019–2021. This investigation provided foundational data for discussing not only the present but also the future role of physical education. In summary, the obesity of fifth- and sixth-grade elementary school students in Korea increased during 2019–2021. This increase may be due to the decrease in students’ physical activity. Male students showed a steeper increase in obesity regardless of grade.

Experts have argued that the amount of physical activity in students has decreased for various reasons and have predicted that it will continue to decrease in the future. Inactive lifestyles at school and increased screen time will lead to a decrease in physical activity, resulting in an increase in obesity and a decrease in physical fitness among students. To prevent this, sufficient discussions must be carried out on how to apply future education, such as EdTech and ICT, to physical education so that students can continue engaging in physical activity even during changes in the social environment. Furthermore, it is necessary to establish measures to protect the health and physical fitness of students in future education.

In particular, schools must make certain physical activities mandatory for students. This can be done through various measures, such as making certain hours of physical education compulsory, applying physical activities during free time at school, and requiring students to participate in a sports club. Unexpected pandemics or rapid social changes make it impossible to guarantee the conditions of students’ environments outside of school. Schools have strived to protect the safety and health of students despite the rapidly changing social trends amid the recent pandemic.

In the end, school and physical education must be at the heart of such measures. More specifically, physical education must play its role even in educational environments such as non-face-to-face distance learning. This will help recover the health and physical fitness of the increasing number of obese students. Now is the time to discuss and prepare these measures.

## Figures and Tables

**Figure 1 healthcare-12-00855-f001:**
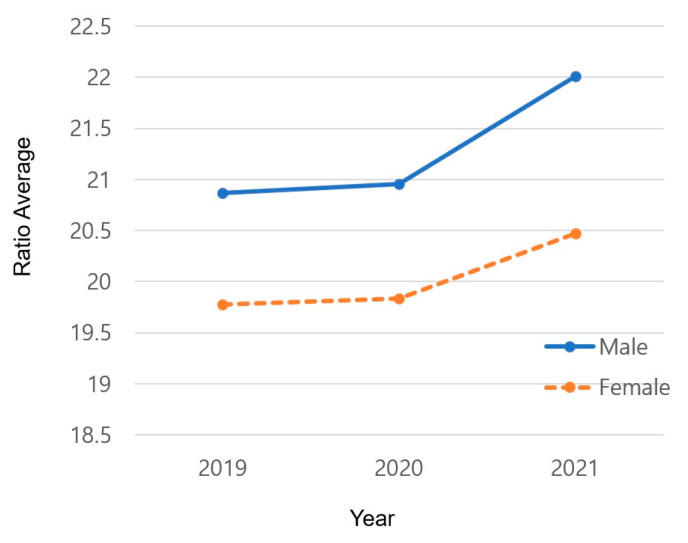
Changes in the yearly average BMI according to gender.

**Figure 2 healthcare-12-00855-f002:**
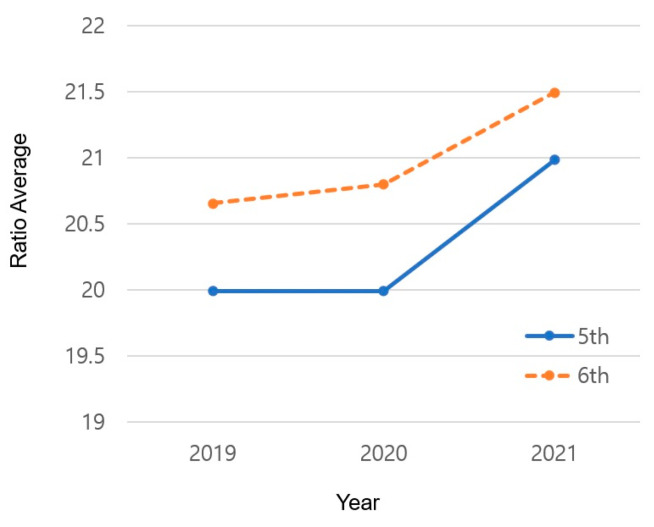
Changes in the yearly BMI average by grade.

**Table 1 healthcare-12-00855-t001:** The general characteristics of the research subjects.

Variable		Number of Students (Proportion)		
		Male	Female	Total
2019	5th grade	33,477 (51.31%)	31,764 (48.69%)	65,241 (100%)
	6th grade	30,980 (52.11%)	28,472 (47.89%)	59,452 (100%)
	Total	64,457 (51.69%)	60,236 (48.31%)	124,693 (100%)
2020	5th grade	31,791 (52.00%)	29,350 (48.00%)	61,141 (100%)
	6th grade	33,429 (51.36%)	31,656 (48.64%)	65,085 (100%)
	Total	65,220 (51.67%)	61,006 (48.33%)	126,226 (100%)
2021	5th grade	29,689 (51.81%)	27,618 (48.19%)	57,307 (100%)
	6th grade	31,481 (52.03%)	29,021 (47.97%)	60,502 (100%)
	Total	61,170 (51.92%)	56,639 (48.08%)	117,809 (100%)

**Table 2 healthcare-12-00855-t002:** Standard table for obesity in Korean elementary schools [22].

Physical Fitness Factor	Item	Male						Female				
Obesity	BMI (kg/m^2^)		Underweight	Normal weight	Overweight	Moderately obese	Severely obese	Underweight	Normal weight	Overweight	Moderately obese	Severely obese
		5th grade	14.5 or below	14.6~21.6	21.7~24.4	24.5~29.9	30 or above	14.2 or below	14.3~20.6	20.7~23.0	23.1~29.9	30 or above
		6th grade	14.8 or below	14.9~22.5	22.6~24.9	25.0~29.9	30 or above	14.6 or below	14.7~21.4	21.5~23.9	24.0~29.9	30 or above

**Table 3 healthcare-12-00855-t003:** Average BMI depending on year, gender, and grade.

Variable	Sum of Squares	Degree of Freedom	Mean Square	F
Year	251.156	1.992	1256.578	362.277 ***
Gender * year	156.292	1.992	78.474	22.592 ***
Grade * year	51.715	1.992	25.966	7.475 **
Gender * grade * year	1.450	1.992	0.727	0.210
Error	33,400.220	9615.638	3.474	

* *p* < 0.05, ** *p* < 0.01, *** *p* < 0.001.

**Table 4 healthcare-12-00855-t004:** Changes in the average BMI by year.

Dependent Variable	Year	Sample Size	Mean	Standard Error
Average BMI	2019	124,693	20.325 ^a^	0.026
	2020	126,226	20.394 ^a^	0.028
	2021	117,809	21.241 ^b^	0.029

Bonferroni: a < b.

**Table 5 healthcare-12-00855-t005:** Yearly BMI average by gender.

Year	Gender	Mean	Standard Error
2019	Male	20.870 ^b^	0.037
	Female	19.780 ^a^	0.037
2020	Male	20.952 ^b^	0.039
	Female	19.837 ^a^	0.039
2021	Male	22.012 ^b^	0.042
	Female	20.470 ^a^	0.042

Bonferroni: a < b.

**Table 6 healthcare-12-00855-t006:** Yearly BMI average by grade.

Year	Grade	Mean	Standard Error
2019	5th grade	19.991 ^a^	0.037
	6th grade	20.659 ^b^	0.037
2020	5th grade	19.992 ^a^	0.039
	6th grade	20.797 ^b^	0.039
2021	5th grade	20.985 ^a^	0.042
	6th grade	21.497 ^b^	0.042

Bonferroni: a < b.

## Data Availability

The data presented in this study are available upon request from the corresponding author. The data were not publicly available because of the protection of personal information.

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
