# Peer review of "Changes in the Average Body Mass Index of Fifth- and Sixth-Grade Korean Elementary School Students: The Role of Physical Education in Student Health"

_healthcare, 2024, doi:10.3390/healthcare12080855_

Round 1

Reviewer 1 Report

Comments and Suggestions for Authors General comments The study investigated the role of physical education on changes in BMI of 5th and 6th Grade Korean Elementary School Students. The study could be rather interesting; however, as clarified by the specific comments below, many points should be addressed more extensively. Specific comments - Improve the scientific English for clarity. Introduction: - Please, delete the first paragraph (lines 34-42). - Lines 60-61. Why the did Authors refer to employment instability in relation to children’s BMI? It is not clear. - Lines 73-79. Please, move this paragraph in the Materials and Methods section. Materials and Methods: - Lines 102-103. BMI is not a measure of body composition, rather it’s a measure of weight status. Please, modify. Items and measurements: - Lines 123-124. What does this sentence mean? Please, rephrase and explain it. - Table 1 and Figure 1 reported the same data. Delete one of them. - Table 2. Standard table for obesity in Korean elementary schools [11]. Please, report the right reference that refers to BMI Korean children standards. Results: - Lines 147-150. Please, move this paragraph into Statistical Analysis section. - Paragraphs 3.2, 3.3 and 3.4. Since these three paragraphs are the explanation of the previous paragraph 3.1, I suggest Authors to rename the paragraphs as 3.1.1, 3.1.2, 3.1.3 for a better clarity. Discussion: - Please, discuss the results of the study in relation to physical education (not only in relation to physical activity). Moreover, what is the role of the school in this panorama? Limitations: - A great limitation of the study is that the results were discussed in relation to a reduction of physical activity level but it was not measured. Moreover, this study was conducted to explore the role of physical education by examining the changes in the BMI of elementary school students but is not clear what is this role of physical education. Comments on the Quality of English Language - Improve the scientific English for clarity.

Author Response

Thank you for considering our research paper. Furthermore, the whole of the manuscript have been 3rd edited by a professional editor such that they are now more readable. We are sure that your advice will increase the completeness of the paper. We revised the paper according to the various opinions you suggested. Once again, We greatly appreciate your review opinion. Please refer to the attached file for specific responses to the review.

Reviewer 2 Report

Comments and Suggestions for Authors

Comments on the Quality of English Language

Author Response

(The authors gave the same response as above.)

Reviewer 3 Report

Comments and Suggestions for Authors

Dear authors,

Thank you for your manuscript. It is an interesting article that investigates a serious problem in an age stage that is usually not considered as much as it should be. Here are some suggestions to improve the quality of your manuscript.

The introduction section is generally well written, however I feel that something is missing. You investigated the role of physical education in student health in the article, but you did not mention it in the introduction. I suggest you insert some of the positive benefits that physical education can provide to elementary school students in general, and to overweight or obese students in particular.

You can refer to studies involving older children or teenagers and state in turn that a few studies have examined this topic in elementary school students.

Materials and Methods

What were the inclusion/exclusion criteria for the study?

Where and by whom were the measurements performed? In-school gyms? In a doctor's office?

By physical education teachers? By specialized doctors?

This information will help the reader better understand the setting and conditions under which the measurements were made.

Please add as much detail as possible to have a clear and in-depth description of the setting.

Out of 49 bibliographic sources, only 21 are after 2015. We are in 2024, so I suggest the introduction of more recent bibliographic sources both in the introduction and the discussions.

In general, the work is well presented and can be published, but it needs to improve these aspects.

Kind regards,

Author Response

(The authors gave the same response as above.)

Round 2

Reviewer 1 Report

Comments and Suggestions for Authors Abstract
Please, better define the aim of the study.
Materials and Methods
Lines 114-115. BMI is not a measure of body cfat, rather it’s a measure of weight status.

Author Response

Thanks to your careful review, we were able to improve the quality of the paper. Thank you very much for this. Attached is the revised manuscript.

Reviewer 2 Report

Comments and Suggestions for Authors

The authors have implemented specific modifications to the manuscript in response to the reviewer's feedback. Following a comprehensive review and verification of both the textual content, figures and tables , the manuscript is now recommended to be accepted for publication.

Author Response

(The authors gave the same response as above.)

Reviewer 3 Report

Comments and Suggestions for Authors

Dear Authors,

thank you for considering my suggestions. The manuscript now looks much better than before. Congratulations on your essential review. Good job!

Author Response

(The authors gave the same response as above.)
